# Sex Differences in Cardiovascular Diseases: A Matter of Estrogens, Ceramides, and Sphingosine 1-Phosphate

**DOI:** 10.3390/ijms23074009

**Published:** 2022-04-04

**Authors:** Beatrice Arosio, Graziamaria Corbi, Sergio Davinelli, Vienna Giordano, Daniela Liccardo, Antonio Rapacciuolo, Alessandro Cannavo

**Affiliations:** 1Department of Clinical Sciences and Community Health, University of Milan, 20122 Milan, Italy; beatrice.arosio@unimi.it; 2Department of Medicine and Health Sciences, University of Molise, 86100 Campobasso, Italy; graziamaria.corbi@unimol.it (G.C.); sergio.davinelli@unimol.it (S.D.); 3Department of Translational Medical Sciences, Federico II University of Naples, 80131 Naples, Italy; vienna.giordano@hotmail.it; 4Department of Neurosciences, Reproductive and Odontostomatological Sciences, Federico II University of Naples, 80131 Naples, Italy; liccardo.daniela@gmail.com; 5Departmento of Advanced Biomedical Sciences, Federico II University of Naples, 80131 Naples, Italy; rapacciu@unina.it

**Keywords:** sex differences, sphingolipids, estrogens, cardiovascular

## Abstract

The medical community recognizes sex-related differences in pathophysiology and cardiovascular disease outcomes (CVD), culminating with heart failure. In general, pre-menopausal women tend to have a better prognosis than men. Explaining why this occurs is not a simple matter. For decades, sex hormones like estrogens (Es) have been identified as one of the leading factors driving these sex differences. Indeed, Es seem protective in women as their decline, during and after menopause, coincides with an increased CV risk and HF development. However, clinical trials demonstrated that E replacement in post-menopause women results in adverse cardiac events and increased risk of breast cancer. Thus, a deeper understanding of E-related mechanisms is needed to provide a vital gateway toward better CVD prevention and treatment in women. Of note, sphingolipids (SLs) and their metabolism are strictly related to E activities. Among the SLs, ceramide and sphingosine 1-phosphate play essential roles in mammalian physiology, particularly in the CV system, and appear differently modulated in males and females. In keeping with this view, here we explore the most recent experimental and clinical observations about the role of E and SL metabolism, emphasizing how these factors impact the CV system.

## 1. Introduction

Sex differences have been observed in cardiovascular diseases (CVD), ultimately leading to heart failure (HF) development and prognosis, but the underlying mechanisms remain unclear [1,2]. In animals and humans with cardiac disease, females display lower mortality, less severe disease phenotype, and faster cardiac repair than their male counterparts [3,4,5,6]. This difference has been primarily attributed to sex hormones such as estrogens (Es) [7,8,9,10]. Es are undeniably protective in women, and their decline during and after menopause coincides with increased CV risk and HF development [11]. Es are known to positively impact cardiac function, remodeling, and metabolism among their functions [8,12,13,14]. For these reasons, E replacement therapy (ERT) has been proposed as a therapeutic strategy to treat post-menopausal women [14,15,16,17,18,19]. However, the use of E as a therapeutic agent in humans is still controversial, as clinical trials demonstrated that ERT in post-menopause women results in adverse cardiac events and increased risk of breast cancer [20,21]. These harmful effects may be attributable to several factors, including the lack of clear comprehension of the biological role of Es and the downstream signaling pathway [18]. Thus, understanding these molecular mechanisms behind E activities and underlying these sex-related differences, in general, will provide a vital gateway toward better CVD prevention and treatment. 

This review focused on sphingolipids (SLs), as previous studies demonstrated how these molecules participate in E downstream signaling [22]. These lipids are derivates of amino alcohol sphingosine (S) and are essential structural components of cell and organelle membranes [23,24]. SLs show biological activities in many critical cellular processes such as proliferation, growth, and survival, thus regulating whole mammalian physiology, including the homeostasis of the CV system [25,26]. Among SLs, ceramide (Cer) and S1-phosphate (S1P) are the two most studied SLs with opposing bioactivities [27]. For this reason, preserving the Cer/S1P rheostat has been proposed as a potential therapeutical strategy to counteract CVDs and other disorders, including cancer [28,29]. Hence, there is a keen interest in exploring how SLs are implicated in CVDs and if their signaling/metabolism is influenced by age- and/or sex.

This review article will show the most clinical and experimental evidence around SLs, sex differences, and Es. Further, we will critically discuss the potential role of these bioactive molecules as diagnostic biomarkers and therapeutic targets for CVDs.

## 2. Sphingolipids Metabolism

In 1874, John Louis William Thudichum discovered the organic base of an enigmatic class of brain lipids called S [30,31]. Indeed, the term S is derived from the mythological creature named Sphinx, known for its love of riddles [30,31]. Later, in 1947, Herbert Carter proposed defining these substances, containing a long-chain amino alcohol backbone (a sphingoid base) attached via an amide linkage to a fatty acyl chain, with the term SLs [32]. 

As shown in Figure 1, the first determining step in the generation of SLs, the de novo synthesis pathway, begins in the cytosolic leaflet of the smooth endoplasmic reticulum (ER) and is characterized by the condensation of amino acid, often a serine, with palmitoyl CoA to form sphinganine by the enzyme serine palmitoyltransferase (SPT) [33,34]. Sphinganine is further acylated into Cers and transferred to the Golgi apparatus via the direct action of the Cer transfer protein (CERT), where it can be modified into sphingomyelins (SMs) and possibly glycosphingolipids [35,36]. Notably, Cers can be regenerated from SMs (hydrolysis pathway) or are deacylated by a family of enzymes called ceramidases to produce S and fatty acids (salvage pathway) [33,34,37]. The S generated can then be phosphorylated by a class of enzymes called S kinases 1 and 2 (SK1/2), localized in the cytosol or associated with specific membrane compartments, leading to the formation of S 1-phosphate (S1P) [38]. S1P is catalyzed by S1P lyase and transformed into fatty aldehydes and ethanolamine phosphate in the final step of SL breakdown [35]. These compounds become substrates for a cascade of enzymatic reactions from which fatty acyl-CoA is obtained. As an alternative, S1P can be dephosphorylated by S1P phosphatase leading back first to S and then to Cer [37,38]. Within a century, the chemical structures of thousands of SLs have been described along with their biological role in eukaryotic cellular processes, including inflammation, cell migration, adhesion, growth, and apoptosis [37,38,39,40,41], and their link with a growing number of diseases [42].

## 3. Ceramide and Sphingosine 1-Phosphate in Cardiovascular Physiology and Disease

Among the SLs identified, Cer and S1P [27,38] are of particular interest in CV system physiology and pathology.

### 3.1. Cer

As described above, Cers are central intermediates in SL metabolism; thus, they are considered one of the most important SLs identified [37,43]. Structurally Cers consist of a fatty acyl of variable chain length, bound to an amino group of S. In mammalian cells, the most abundant Cers contain long (C16–18) and very long (C20–24) acyl chains. However, those with longer acyl chains (C26–36) have also been found in male germ cells and epidermal keratinocytes [44]. Together with SM, Cers are essential constituents of the cell membrane, participating in cell signaling regulating the localization and activation of membrane-associated receptors [45]. Cers are involved in several biological activities that influence the pathophysiology of CVDs, with effects on apoptosis, oxidative stress, inflammation, endothelial dysfunction, insulin resistance, and lipotoxicity [46,47]. For this reason, Cer modulation has been correlated to various pathological conditions, particularly those affecting the CV system. In this regard, it is worth recalling that Cer, and all SLs in general, are perfect biomarkers because they can be identified in tissues and biological fluids and quantified by several approaches commonly used in clinical routines such as immunoassays and enzymatic assays (measuring diacylglycerol and Cer kinase activity) [48,49,50] or by thin-layer chromatography [48]. While TLC allows the separation and characterization of Cers by retention factor compared with known standards [51], the immunoassays and enzymatic assays only detect the total Cer content, leaving unrevealed the saturated and unsaturated acyl chains. Thus, more complex and expensive approaches to measure these SLs, like nanoscale liquid chromatography (LC) coupled to nanoelectrospray ionization mass spectrometry (MS), have also been used for their high specificity and sensitivity [50]. Notably, over the last decades, all these approaches helped to reveal how distinct plasma Cer species and ratios are significant predictors of CVD. Indeed, several pre-clinical and clinical accumulating evidence suggests that Cer functions are chain-length specific. For instance, Turpin et al. [52] showed a relative increase in long-chain species (C16:0) but not in the very long-chain (C24:0/24:1) in adipose tissue of obese humans and mice, and also correlated with insulin resistance. These results were consistent with previous results from Raichur and coworkers showing that mice with upregulation of C16:0 Cers, developed hepatosteatosis and insulin resistance [53]. Analogously relevant is the observation that long-chain Cers are pro-apoptotic, whereas those with a very-long-chain are prosurvival [54]. Thus, it is imperative to dose all the species separately to perform a correct association. 

In one of the first studies performed on humans, de Mello et al. [55] measured plasma Cer species in patients with coronary heart disease (CHD). Interestingly, the authors demonstrated that both total and Cer subspecies d18:1/C23:0 and d18:1/C24:1 were all associated, with no difference among them with regard to insulin resistance and increased inflammatory status of patients with established CHD. Similar results were observed by Spijkers et al. [56] that through MS lipidomic analysis observed augmented total Cer levels, due to increases in C16:0, C22:0, C24:1, and C24:0, in blood plasma from spontaneously hypertensive rats (SHRs) and in humans with essential hypertension.

Subsequently, Pan et al. [57] measured plasma Cer levels in patients with unstable angina pectoris and acute myocardial infarction and demonstrated a significant elevation in these patients compared to those in the control group. Similarly, results were obtained by Yu and coworkers [58] that in a cohort of Chinese patients with chronic heart failure (CHF) analyzed plasma Cers (as a sum of all subtypes). Interestingly, the authors observed that circulating levels of Cers (total) were significantly increased in CHF patients and correlated with severity and reduced left ventricular function. Thus, Cer is an independent risk factor of mortality due to CHF.

However, there are some discrepancies in the association of different types of Cers to CVD risk. For example, Havulinna et al. [59], in a large-scale prospective study (FINRISK), differentiated the prognostic value of each subspecies in predicting major adverse cardiovascular events (MACEs) among a population of apparently healthy individuals and in patients with pre-existing disease. These authors reported that while Cer (d18:1/18:0) was able to predict MACEs in healthy subjects, Cer (d18:1/16:0) and (d18:1/24:1) [60], correlated more with new events in patients with coronary artery disease and acute coronary syndrome. This study supported the notion that Cers are responsible for developing atherosclerotic vascular disease and the progression of vascular adverse events in both humans and rodents [61]. In this regard, a recent study from Pan et al. [62], further explored the role of Cers in atherosclerosis, demonstrating how Cers (d18:1/16:0), (d18:1/18:0), (d18:1/24:1) and (d18:1/24:0) were all predictors of plaque rupture. Importantly, Cers can also be a direct predictor of cardiac dysfunction. In this context, data from the Framingham Offspring Study showed that increased Cer species, in particular Cer ratio (C16:0/C24:0) in plasma, was associated with increased cardiac remodeling and dysfunction in humans [63]. In line with these data, Targher et al. [64], observed that increased Cer (d18:1/24:0) was associated with a greater risk of cardiovascular mortality in patients with CHF. Further to these reports, Lemaitre et al. [65] reported that higher levels of plasma Cer and SM species containing a palmitic acid (C16) are associated with a higher risk of HF, while Cer (C22) species with longer saturated fatty acids are associated with lower risk. 

Hence, apart from being possible therapeutic targets, these studies support Cers more as promising biomarkers for CVDs. For this reason, a “Cer score” has been established [66], that is to be slowly introduced into clinical practice [60].

### 3.2. S1P

In the “salvage” or catabolic pathway, Cer is broken down by ceramidases to generate S, which, in turn, is phosphorylated by SK1 and 2 leading to the formation of S1P [38]. The bioactive S1P is a lipid mediator exported out of the cells, mainly by red blood cells (RBCs), platelets, fibroblasts, vascular smooth muscle cells (VCMCs), endothelial cells (ECs), and cardiomyocytes [67,68,69,70,71], determining the so-called “inside-out” signaling [72]. In addition, extracellular SK1, released from these cells, participates in S1P generation [70]. Importantly, S1P exerts a crucial role in many cellular signaling cascades and pathological processes. These activities are mediated by specific G protein-coupled receptors (GPCRs), called S1PRs [73,74,75,76]. Five S1PRs (S1PR1-5) have been identified so far, differing in tissue and cell expression [38], with S1PR1-3 primarily expressed in the cardiovascular system [77,78,79,80]. Notably, in ECs and cardiomyocytes, S1PR1 is the most expressed S1PR subtype. In contrast, S1PR3 is much more represented in fibroblasts than S1PR1 and S1PR2 [81]. It is worth noting that the specific G protein coupled to each S1PR dictates the effects induced by S1P. For instance, G protein alpha subunit (Gαi) signaling leads to the inhibition of the adenylate cyclase and a reduction of the second messenger cyclic adenosine 3′,5′-monophosphate (cAMP) [82]. Moreover, Gαi can activate PKCα and 𝜀, modulating calcium uptake [83,84]. Conversely, Gαq activation induces phospholipase C (PLC) that, in turn, hydrolyzes phosphatidylinositol 4,5-bisphosphate (PIP2) to produce diacylglycerol (DAG) and inositol trisphosphate (IP3) [82]. Finally, the Rho guanine nucleotide exchange factor (Rho-GEF) is the main effector of Gα13 and Gα12, activating downstream low molecular Rho GTPases [85]. 

Through the activation of S1PRs, S1P plays a pivotal role in developing the vasculature and its stabilization [78,86]. For instance, in vascular ECs, S1P via direct binding of S1PR1 and 3, protects the cells against pro-apoptotic stimuli and promotes proliferation and migration, playing a vital role in angiogenesis [85,87,88]. Conversely, the activation of S1PR2 inhibits EC function.

In VSMCs, which mainly express S1PR2 and S1PR3 [88,89], S1P enhances the cell migratory capacity through direct binding S1PR3. Conversely, when S1P binds S1PR2 signaling, it impairs VSMC functionality [88,89,90].

Analogously important is the role of S1P in cardiomyocytes and cardiac fibroblasts. Indeed, several pieces of evidence demonstrate that, opposite to Cer, which induces apoptosis in cardiomyocytes, S1P possesses cardioprotective effects [38]. For instance, stimulation with S1P of cultured rat neonatal cardiomyocytes prevents ischemia-induced cell death [91,92]. Similarly, in vivo, Bandhuvula et al. [93] demonstrated that mice lacking the enzyme S lyase, which degrades S1P, were protected against ischemia/reperfusion (IR) injury. Importantly, these authors observed that heterozygous S lyase knockout mice presented with elevated S1P levels, consequent smaller infarct size, and better functional recovery after I/R than their controls. Similarly, we recently reported that S1P serum levels were significantly reduced in post-ischemic heart failure (HF) WT mice and normalization of circulating S1P levels, through exogenous application of S1P, positively impacted cardiac function, and remodeling [78]. 

Notably, in fibroblasts, S1P acts as a positive regulator of fibroblast function, enhancing their migration and proliferation, and these effects are associated with the activation of the downstream S1PR2 signaling pathway, which includes the mitogen-activated protein kinase (MAPK) ERK and Rho [94,95]. 

In light of this evidence, S1P levels in blood plasma and serum have been evaluated as predictive of the presence and severity of CVD, as in the case of obstructive coronary artery disease (CAD), myocardial infarction (MI), atherosclerosis, and HF in humans [96,97,98,99,100]. Accordingly, Egom et al. reported that S1P soon increases after myocardial ischemia, caused by percutaneous coronary intervention, but gradually returns to baseline levels in 12 h [101]. Importantly, these authors demonstrated that this rapid increase of S1P is compensatory and provides cardioprotection against ischemia [101]. In line with these results, Klyachkin and colleagues showed that a rise in S1P in response to MI is a good strategy to trigger the mobilization of bone marrow (BM)-derived stem/progenitor cells (BMSPCs) and enhance the recovery of the ischemic myocardium [102]. Further, Knapp and coworkers [103] reported a reduction in circulating S1P levels in patients with MI compared to controls, supporting the notion that a reduction in S1P levels was associated with a worse outcome. In this regard, in mouse and rat models of post-ischemic HF, we recently demonstrated that either cardiac or circulating levels of S1P are reduced compared to non-ischemic controls [77,78]. These data were confirmed in humans by Polzin et al. [104], demonstrating that patients with severely reduced left ventricular (LV) ejection fraction (LVEF < 40%) have lower plasma S1P levels than those with mildly reduced LVEF (LVEF > 40%). These authors observed that levels of plasma S1P of NYHA class III and IV patients were lower than those of NYHA class I and II [104]. Therefore they concluded that reduced plasma S1P levels were negatively associated with LVEF and clinical signs of heart failure. In line with the concept that a reduction in S1P levels is noxious in the CV system, Soltau and colleagues [105] reported that a decline in serum-S1P levels was inversely associated with peripheral artery disease (PAD) carotid stenosis (CS) in humans. These authors demonstrated that S1P was a more accurate circulating marker for predicting the presence of PAD and CS than HDL. However, it is paramount that a significant amount (∼65%) of S1P in the human plasma is carried primarily by the apolipoprotein M (ApoM) subfraction of the high-density lipoprotein (HDL) while the remaining portion (35%) is bound to albumin [106]. Therefore, S1P is seemingly a direct mediator of many cardiovascular effects attributed to HDL. For instance, HDL-mediated vasorelaxation and prosurvival actions on the endothelium and cardiomyocytes have been attributed to S1P activity [107]. Moreover, HDL declines in diseases like atherosclerosis, CAD, MI, and diabetes, appearing to drive the reduced concentration of S1P in the plasma [97]. Indeed, Sattler and colleagues reported that HDL-S1P levels in MI and stable CAD patients were lower than in controls. However, when plasma S1P levels of these patients were normalized to HDL levels, these resulted in being augmented compared to healthy subjects [97]. Finally, one of the last aspects to analyze is the elevation in plasma S1P in obesity and diabetes, representing two major CVD risk factors [108]. Notably, Kowalski et al. reported that in both humans and rodent obesity, increased plasma S1P levels correlated with adiposity and insulin resistance [109]. Ito and colleagues observed that plasma S1P levels were increased in obese patients compared with those detected in non-obese and lean individuals [110]. The significance of the elevation of S1P in the plasma of obese patients remains still under debate. Indeed, studies suggest that S1P metabolism could be either negatively or positively linked to the pathology and the onset of insulin resistance and diabetes.

## 4. Estrogens and CVD

It is well recognized that menopausal women undergo CVD more than men of the same age [111]. The causes for this sex/gender difference remain a subject of debate but seem likely to be linked to the beneficial effects of sex hormones, particularly estrogens (Es) [7,8,9,10]. As age progresses, during and after menopause, there is a cessation of ovarian follicle development followed by dramatic declines in E levels, especially 17β-estradiol (E2), the most potent and active metabolite in humans. Therefore, it has fueled interest in the possibility that E replacement therapy (ERT) can protect against CVD among post-menopausal women [14,15,16,17,18,19]. Es mediates protection within the CV system, and this has been essentially related to the effects exerted at vascular and cardiomyocyte levels. There are two established E receptors (ERα and β) localized at the plasma membrane, in the cytoplasm, and in the nucleus of these cells [8,112,113]. Moreover, a third membrane-associated ER called G-protein-coupled E receptor (GPER or GPR30) has been identified [114] and has been shown to contribute to E2 mediated effects [115]. For instance, in ECs, E2 activates all these receptors with subsequent induction of the phosphoinositide 3-kinase (PI3K)/Akt and endothelial nitric oxide synthase (eNOS) intracellular signaling pathway, culminating in the production of nitric oxide (NO). Importantly, this gaso-transmitter has a vital role in the cardiovascular system [116,117] and mediates E2 effects on the endothelium in vitro and in vivo [118,119,120,121,122]. In particular, E2 has been shown to protect the arteries from high pressure-induced damage, typical of hypertension [113]. Moreover, Nakamura and coworkers [123], in line with previous studies in vitro by Okubo et al. [124], and Seeger et coworkers [125] demonstrated that E2 also exerts an anti-atherogenic effect. This effect appears to be mainly dependent on the inhibition of VSMC proliferation, which, as reported by Ortmann et al. [126], is strictly related to ERα and β activation, but not to GPER. Further, E2 prevents abnormal monocyte adhesion and transmigration in the vasculature, another critical event in the development and progression of atherosclerosis [127,128,129]. In this regard, Kurokawa et al., in line with previous studies, reported in vivo in ovariectomized (OVX) Sprague–Dawley rats that E2 and its endogenous metabolite 2-methoxyestradiol prevented monocyte adhesion to the aortic endothelium, thus mediating an anti-atherosclerotic effect [130]. Further, Simoncini et al. demonstrated that E2 inhibited VCAM-1 expression and blocked the adhesiveness of monocytes to human saphenous vein ECs, after lipopolysaccharide (LPS) treatment [131]. Notably, LPS is an outer membrane component of Gram-negative bacteria and is considered one of the most critical factors responsible for vascular inflammation [132,133,134]. In this regard, LPS constitutes a decisive risk factor for the development of atherosclerosis [133,134,135]. Notably, LPS activates a cluster of cells within the CV system, such as monocytes, vascular ECs, and VSMCs [136]. Among the factors secreted by these cells in response to LPS, the monocyte chemoattractant protein-1 (MCP-1) is an inflammatory cytokine that regulates the migration and infiltration of monocytes/macrophages [137]. Moreover, this chemokine stimulates the recruitment and enhancement of VSMC migration towards the subendothelial area, accelerating atherosclerosis progression [137,138,139,140]. Importantly, Jiang et al. [138] demonstrated that E2 prevented the elevation of MCP-1 in vitro in VSMCs following LPS stimulation. 

The protective effects of Es on the CV system also involve direct actions on cardiomyocytes. These cells express both subtypes of ER, ERα, and β, with significantly higher levels of ERα [141,142,143]. Notably, through ERα activation, E2 has been shown to regulate cardiac bioenergetics. For instance, Chen et al. [144] showed E2 supplementation in an OVX mouse model of a human hypertrophic cardiomyopathy mutation (cTnT-Q92 mice) resulted in increased ATP levels and mitochondrial respiratory function with preservation of myocardial function. Further, Arias-Loza and coworkers demonstrated that the E2/ERα system entirely prevented the decrease of cardiac glucose uptake observed in OVX mice. Two years later, Devanathan et al. [145] reported, for the first time, that cardiomyocyte-specific deletion of ERα impacted metabolic gene expression in cardiomyocytes in a sex-dependent manner. 

Interestingly, anti-apoptotic and pro-survival effects have also been attributed to Es. In this regard, Pelzer and colleagues reported that E2 interferes with NF-kappaB activity in cardiomyocytes, thus inhibiting apoptosis in response to staurosporine. Of note, several independent reports demonstrated that part of these anti-apoptotic effects attributed to E2 was strictly dependent on ERβ activation. In this regard, Fliegner et al. [146] reported that ERβ mice underwent pressure overload via transverse aortic constriction and exhibited augmented expression of pro-apoptotic genes with a consequent increase in cardiomyocyte apoptosis and cardiac fibrosis. In line with these data, one year later, Cao and coworkers [147] demonstrated in vivo in mice undergoing surgical-induced MI that E2 treatment resulted in a reduced prevalence of cardiac rupture, associated with a decline in matrix metalloproteinase 9 (MMP-9) activation and enhanced expression of the anti-apoptotic gene Bcl-2, compared to their controls (MI mice treated with placebo). Further, in vitro, these authors reported that H9c2 cells subjected to ischemia-reperfusion injury, with E2 treatment via Erβ, mediated cytoprotective effects via Akt-Bcl-2 signaling pathway activation. Similar results were observed in vitro in neonatal rat ventricular myocytes (NRVMs) by Hsieh et al. [148] and subsequently by Lin and colleagues [149] who demonstrated how the E2/ERβ system abolished the noxious effects of isoproterenol stimulation in vitro in the H9c2 cells (a surrogate of cardiomyocytes), blocking the apoptotic response through the enhancement of PI3K/Akt/MDM2 signaling pathway.

However, a role for GPER activation downstream of E2 anti-apoptotic signal in cardiomyocytes has also been demonstrated. Indeed, this receptor is highly expressed in cardiomyocytes [150] and as reported by Li et al. [151] provides a cardioprotective effect following ischemia-reperfusion in H9c2 cells via enhancement of Bcl-2 expression and reduced Bax levels with a consequent impaired apoptotic response. Interestingly, these authors demonstrated that GPER activation was also responsible for increased activation of the antioxidant molecule superoxide dismutase, supporting for E2 also an antioxidant role. Indeed, as previously reported by us [152], GPER activation can prevent oxidative stress in NRVMs. This effect was partly attributed to the ability of E2/GPER signaling to inhibit the phosphorylation of the G-protein-coupled receptor kinase 2 (GRK2). Importantly, GRK2, when phosphorylated at the serine 670, translocates from the cytosol to the mitochondria, thus activating pro-apoptotic signaling and increasing oxidative stress [152,153,154]. This effect is particularly pronounced in the presence of the hormone aldosterone secreted by the adrenal cortex, that when present at high levels, induces cardiotoxic effects with consequent increased reactive oxygen species (ROS) generation and subsequent cell death. Significantly, GRK2 can block GPER protective effects in cardiomyocytes [155]. Indeed, like other GPCRs [156], GPER exerts part of its anti-apoptotic effects via transactivation of the epidermal growth factor receptor (EGFR) and, as shown by Maning et al. [155], GRK2 phosphorylates and desensitizes GPER, thus abolishing this beneficial signaling pathway.

## 5. Estrogens and Sphingolipids

ERT was implemented in clinical practice more than 50 years ago [157,158]. Interestingly, this therapy’s introduction was mainly based on the observation that E reduction in post-menopausal women or those who underwent surgical removal of the ovaries (hysterectomized) was associated with an increased risk of CVD. Moreover, studies showing the benefits of ERT in preventing osteoporosis further encouraged its application [157,158]. However, despite a large-scale observational study, the nurses’ health study supported ERT for its potential benefit against CVD, a concurrent study (the Framingham study) reported no advantages in terms of mortality from all causes and CVD between E users and nonusers. Further, data from the Women’s Health Initiative (WHI), raised several concerns regarding the potential increased risks of breast cancer, stroke, CHD, and pulmonary embolism in women with an intact uterus treated with Es plus progestin [159,160]. Two years later, results from another trial analyzing the effects of E alone (ET) in hysterectomized women reported an association between Es and the increased risk of stroke with no benefit for CHD prevention [157,158]. Therefore, the enthusiasm for ERT usage in post-menopausal women has been significantly reduced over the years. Of note, one of the significant causal factors associated with the failure of ERT is probably the compound used for these therapies. For instance, most of the trials, including those cited above, analyzed the effects of conjugated equine estrogens (CEE), which contains ten or more biologically active Es [161,162] instead of E2 alone. Indeed, E2 is the major naturally occurring E in women and, as discussed above, almost all of the experimental data in the literature undeniably attest its protective role in CVDs. Importantly, such a hypothesis was corroborated by an observational study from Smith et al. [163], demonstrating that CEE use was associated with a higher risk of incident venous thrombosis and possibly MI than E2 alone. However, part of the side effects of ERT may also be attributed to the lack of clear comprehension of the biological role of Es and their downstream signaling pathway. In this regard, it is worth noting that several studies have demonstrated a significant correlation between SLs, by means of Cer and S1P, and Es in breast cancer development [164,165]. In this regard, Schiffmann and colleagues [166] analyzed Cer levels in 43 malignant breast tumors and compared them with normal tissues. Interestingly, they found that total Cer levels in malignant tumor tissue samples were significantly increased compared with those found in normal tissues. Moreover, these authors demonstrated that Cer (18:0) and Cer (20:0) were significantly higher in ER-positive tumor tissues than in ER-negative ones. Further, numerous reports from 2002 supported the importance of the SK1/S1P system downstream E2 in breast cancer development and progression. Therefore, this confirms, in part, the relationship between SLs and the side effects associated with ERT in women. Nevertheless, these observations widened the attention on the Es and SL relationship in the CV system and disease. Indeed, as discussed in the previous paragraphs, Cer and S1P regulate CV homeostasis oppositely. Thus, the CV system’s Es-dependent beneficial vs. harmful effects are plausibly related to the modulation of one of these two SLs. However, it is important to underline that cancer cells cannot fully reflect the characteristics of normal cells. Therefore, an interesting question is whether Es regulate SLs and how they impact the CV system. Interestingly, in premenopausal women, Es has been shown to lower LDL and raise HDL, contributing to the observed cardiovascular benefit [167]. Conversely, in post-menopausal women, the decline in Es levels results in an increase in LDL and a drop in HDL with a rapid progression towards atherosclerosis and coronary artery disease. Parallel to these findings, Guo et al. [168] reported that plasma S1P levels were significantly higher in women aged 19–55 than men of the same age and decreased in postmenopausal women. Moreover, these authors demonstrated that in ECs, E2 stimulation induced a direct activation of SK1 with consequent production and release of S1P, suggesting the vasculoprotective effects of the E2/S1P system. In 2019, Vozella and coworkers [169], studied a cohort of 84 women (19 to 80 years of age) and observed that plasma Cer (d18:1/24:1) levels positively correlated with age whereas negatively correlated with plasma E2. In line with these observations, Li et al. analyzed the effects of E depletion in mice on SLs and CV functionality. Interestingly, these authors observed that OVX mice presented increased blood systolic, diastolic, and pulse pressure compared to the group sham (non-OVX). Notably, depletion of Es resulted in a significant increase in total Cer levels while S1P decreased significantly, further strengthening the role of Es as a bridge between SLs and CV system homeostasis. The importance of the aging process on SL release in women has been recently reported by our group [170]. In particular, we characterized for the first time the serum SL profile in adults (35–37 years old) vs. aged (75–77 years old) vs. long-lived (centenarians; 105–107 years old) women and in accordance with previous results we observed that Cer levels increased with age. Next, analyzing the levels of S1P we also observed that both centenarians and aged women presented higher levels than adults. Whether and how SLs, affect positively or negatively, the physiology of aged and centenarians was not assessed. Anyway, analyzing the results from this study, we postulated that in aging, Cers are generated through the de novo pathway. In contrast, in centenarians, Cers are derived from SM degradation (hydrolysis pathway) then, they are probably converted into glycosphingolipids and subsequently metabolized to gangliosides [171]. Therefore, our study supports the idea that maintaining a cellular balance between Cer/S1P (rheostat) contributes to the control of CV system functionality and the prevention of several pathological conditions typical of the post-menopausal period (i.e., cancer and neurodegeneration) [172,173,174,175] and typically associated with the alteration of SL metabolism [28,29,176,177]. Importantly, E2 administration is undoubtedly a strategy to induce vascular protection via preservation of the Cer/S1P rheostat (Figure 2). However, as discussed above, the risk associated with ERT outweighs the benefits, thus the modern perspective can be to use plant-based therapies like phytoEs (PhEs) and their derivatives [178] (Figure 2). These polyphenolic estrogenic compounds of plant origin are classified in four main classes (isoflavones, lignans, coumestans, and stilbenes) and possess E-like activity binding both ERα and ERβ [179]. And as per previous studies, appear to increase both Cer and S1P in vitro in HepG2 cells and in primary rat hepatocytes [179,180,181]. Importantly, over the last years, some experimental and clinical data have demonstrated the potential of these E-like molecules to address several conditions associated with menopause, including CVDs [182,183]. Indeed, we recently demonstrated that dietary supplementation with two PhEs, particularly equol and resveratrol, can improve menopause-related quality of life in healthy women [178], whereas Lee et al. [184] analyzed the effects of PhEs in patients with hypertension. In particular, these authors observed a positive association between plasma PhE concentration (mainly equol and enterolactone) and the prevention of hypertension. Other studies, including the report from Hammad and colleagues [185] demonstrated that the usage of resveratrol during the post-menopausal period is a valid and safer alternative to Es to prevent vascular calcification. Finally, Wolters et al. [186] defined the effects of PhE supplementation in postmenopausal women in a metanalysis of randomized controlled trials. Interestingly, this study demonstrated that PhE supplementation was associated with a decrease in serum total cholesterol, LDL, triglycerides, and apolipoprotein B, increased serum apolipoprotein A-1, and improved several parameters of endothelial function. Therefore, all these studies support the role of PhEs in menopausal CV health and suggest a potential implication of SLs downstream of their mechanism of action.

## 6. Conclusions and Perspectives

Sex-related differences in CVD risk, disease presentation, response to therapy, and prognosis have been well characterized, but the underlying mechanisms and their implications for clinical practice are poorly understood [187]. This is primarily due to the scarcity of sex-specific studies and concerns about tailoring clinical care individually for women and men. Secondarily, most studies did not fully assess all cardiovascular health factors. Despite these gaps, several specific biomarkers have been identified, helping to unveil and clarify the age-based and sex variation in CV outcomes and phenotypes over the past two decades. Moreover, the analysis of behavioral risk factors for CVDs and metabolic disorders has undoubtedly helped clarify the differences between the genders [188,189]. For instance, it has been demonstrated that men are more likely to have poor CV health than women [189], and this has been in part supported by the role of antioxidant enzymes like superoxide dismutase (SOD), glutathione peroxidase, NADPH-oxidase, which are differentially expressed in males and females [190].

Further, the role of dietary intake related to gender has also been reported. In particular, it has been observed that women present with a higher composite dietary antioxidant index (CDAI), constructed based on the intake of zinc, selenium, vitamin A, vitamin C, vitamin E, and carotenoid compared to men. Interestingly, the CDAI in women negatively correlates with CV risk factors and is positively associated with HDL. The specific mechanism behind these differences has not been thoroughly investigated. However, as seen throughout this review, clinical and pre-clinical studies suggest a pivotal role of SL metabolism and levels, which per previous experimental studies in vitro and in vivo in rats are modulated by Es and also by the aforementioned antioxidant molecules [191,192], as predictors of the CV risk in a sex-specific manner. Notably, the role of these molecules is constructive, particularly when correlated with E and PhE levels. Cers and S1P, are not exclusively structural components of plasma membranes but, as discussed in this review, exhibit biological activities outside the cells in an autocrine and paracrine manner. Although they act oppositely, a balance between Cer and S1P levels is crucial for the homeostasis of the CV system. Of course, future studies are needed to investigate whether the observed changes in Cer/S1P concentrations by age and sex reflect other processes. However, as emerged in this review dissecting the role of all these molecules will provide the missing link between postmenopausal hypoestrogenism and increased CV risk, optimizing diagnostic and therapeutic strategies for treating CVDs selectively in men and women and expanding the concept of “sex/gender CV medicine” [193]. What is analogously important will be to shed light on innovative and alternative therapies able to enhance Cer/S1P beneficial effects providing women at high risk of cardiovascular morbidity and mortality with more options for preventative treatment. In this regard, it will be crucial to establish the role of dietary intake in increasing the consumption of nutrients rich in PhEs and antioxidants in postmenopausal women.

## Figures and Tables

**Figure 1 ijms-23-04009-f001:**
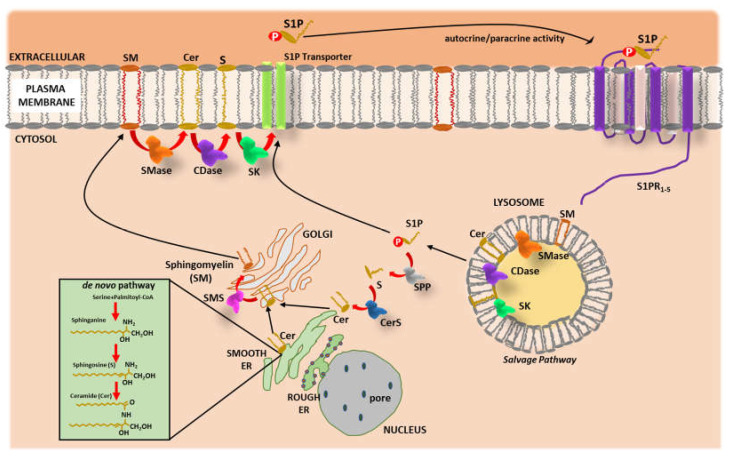
Schematic representation of sphingolipid metabolism. The de novo synthesis pathway begins in the cytosolic leaflet of the smooth endoplasmic reticulum (ER). It starts with the condensation of palmitoyl coenzyme A (CoA) and serine, followed by several enzymatic reactions (red arrows) leading to the formation of sphinganine, sphingosine (S) and then ceramide (Cer). Once generated, Cer is transported to the Golgi apparatus where it is converted to sphingomyelin (SM) by SM synthase (SMS). SM is then transported to the plasma membrane where under specific condition/stimulation, SM can be reconverted in Cer by sphingomyelinase (SMase) that is further transformed into S and with the help of sphingosine kinases (SKs) is phosphorylated, thus leading to the generation of the bioactive lipid S1-phosphate (S1P) that is transported outside the membrane. Of note, the production of S1P is also possible in the lysosomes (salvage pathway) where SM is converted into Cer and then S. The S1P generated in this pathway can be either transported outside the cells or is reconverted into S by S1P phosphatase (SPP). S is finally transformed into Cer by Cer synthase.

**Figure 2 ijms-23-04009-f002:**
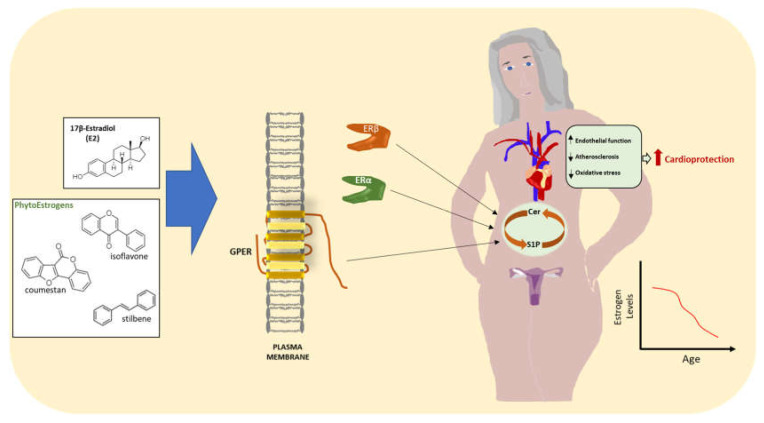
Potential effects of 17β-estradiol (E2) or phytoestrogen (PhE) therapy in postmenopausal women. E2 and PhEs bind to ERs and GPER, thus preserving Cer/S1P rheostat with a consequent beneficial effect on the cardiovascular system. The chemical structures were produced on https://chemdrawdirect.perkinelmer.cloud/js/sample/index.html, accessed on 1 January 2022.

## Data Availability

Not applicable.

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
