# Peer review of "Sex Differences in Cardiovascular Diseases: A Matter of Estrogens, Ceramides, and Sphingosine 1-Phosphate"

_ijms, 2022, doi:10.3390/ijms23074009_

Round 1
Reviewer 1 Report
Author has reported the most recent experimental and clinical observations about the role of Es and SLs' metabolism, emphasizing how these factors impact the CV system. However, dissecting the role of these molecules will provide the missing link
between post-menopausal hypoestrogenism and increased CV risk;.optimizing both diagnostic and therapeutic strategies for treating CVDs selectively in men and women, and expanding the concept of "sex/gender CV medicine. This work is interesting and highly citable. Although the article has scientific rigor, minor flows need to be corrected before publication.
The author may add a recent references in the introduction section
Grammatical and typo errors are found
Provide some future paths of this study in conclusion.
Author Response
REVIEWER 1:
Author has reported the most recent experimental and clinical observations about the role of Es and SLs' metabolism, emphasizing how these factors impact the CV system. However, dissecting the role of these molecules will provide the missing link between post-menopausal hypoestrogenism and increased CV risk; optimizing both diagnostic and therapeutic strategies for treating CVDs selectively in men and women, and expanding the concept of "sex/gender CV medicine. This work is interesting and highly citable. Although the article has scientific rigor, minor flows need to be corrected before publication.
Reply: We thank the Reviewer for His/Her constructive feedback and comments on our manuscript. We have revised the manuscript to address all the points raised. We trust that the responses and corresponding changes made to the revised manuscript, as detailed below, have satisfactorily addressed all the issues and that our revised manuscript will now be deemed suitable for publication in the International Journal of Molecular Sciences.
1) The author may add a recent references in the introduction section
Reply: We thank the Reviewer for raising this concern. As indicated, we updated the reference list. We added more recent references in the Introduction section to describe the implication of ERT in post-menopausal women (Gersh et al. Heart 2021 Prabakaran et al. Endocr Pract. 2022; Pan M et al. Biosci Trends. 2022). Please see page 1, lines 42-44)
2) Grammatical and typo errors are found:
Reply: As indicated, we have revised our manuscript overall. Thanks
3) Provide some future paths of this study in conclusion:
Reply: This is another good point. We have included a discussion about future paths in the Conclusion section. Thanks.
Reviewer 2 Report
In my opinion, this is an important article in this field of research, summarizing current evidence on the role of sphingolipids and estrogens in dertiming sex-difference in cardiovascular health and disease.
In general, the author discussed that sex-differences are mainly attributable to the effect of estrogens and sphingolipids metabolism. I surely agree with this statement, even if from my point of view, most studies did not assess other factors affective cardiovascular health. For example, it has been widely demonstrated how behavioral risk factors for CVDs and metabolic disorders exhibit differences by gender (please consider for example DOI: 10.3389/fpubh.2020.00108; doi: 10.1177/2047487319834875). Moreover, it has been also demonstrated that some of these factors differentially act on male or female individual (please consider for example DOI: 10.1016/j.freeradbiomed.2018.12.018)
For these reasons, I would suggest to include a section (or to add some text in an existing paragraph), discussing on these points.
Minor points
Please double check the text for some typos
Author Response
REVIEWER 2:
In my opinion, this is an important article in this field of research, summarizing current evidence on the role of sphingolipids and estrogens in determining sex-difference in cardiovascular health and disease.
Reply: We thank the Reviewer for His/Her constructive feedback and comments on our manuscript. We have revised the manuscript to address all the points raised. We trust that the responses and corresponding changes made to the revised manuscript, as detailed below, have satisfactorily addressed all the issues and that our revised manuscript will now be deemed suitable for publication in the International Journal of Molecular Sciences.
1) In general, the author discussed that sex-differences are mainly attributable to the effect of estrogens and sphingolipids metabolism. I surely agree with this statement, even if from my point of view, most studies did not assess other factors affective cardiovascular health. For example, it has been widely demonstrated how behavioral risk factors for CVDs and metabolic disorders exhibit differences by gender (please consider for example DOI: 10.3389/fpubh.2020.00108; doi: 10.1177/2047487319834875). Moreover, it has been also demonstrated that some of these factors differentially act on male or female individual (please consider for example DOI: 10.1016/j.freeradbiomed.2018.12.018)
Reply: As indicated, we have included and discussed the mentioned above studies in the revised version of our manuscript. Please see the Conclusion section. Thanks
2) For these reasons, I would suggest to include a section (or to add some text in an existing paragraph), discussing on these points.
Reply: We thank you again for raising this important issue. As replied above, we have discussed these points in the Conclusion Section.
MINOR POINTS
1) Please double check the text for some typos
Reply: We have revised the manuscript accordingly. Thanks